# Broccoli-Derived Glucoraphanin Activates AMPK/PGC1α/NRF2 Pathway and Ameliorates Dextran-Sulphate-Sodium-Induced Colitis in Mice

**DOI:** 10.3390/antiox11122404

**Published:** 2022-12-04

**Authors:** Qiyu Tian, Zhixin Xu, Qi Sun, Alejandro Bravo Iniguez, Min Du, Mei-Jun Zhu

**Affiliations:** 1School of Food Science, Washington State University, Pullman, WA 99164, USA; 2Department of Animal Sciences, Washington State University, Pullman, WA 99164, USA

**Keywords:** glucoraphanin, inflammatory bowel diseases, DSS, NRF2, oxidative stress, mitochondrial homeostasis

## Abstract

As the prevalence of inflammatory bowel diseases (IBD) rises, the etiology of IBD draws increasing attention. Glucoraphanin (GRP), enriched in cruciferous vegetables, is a precursor of sulforaphane, known to have anti-inflammatory and antioxidative effects. We hypothesized that dietary GRP supplementation can prevent mitochondrial dysfunction and oxidative stress in an acute colitis mouse model induced by dextran sulfate sodium (DSS). Eight-week-old mice were fed a regular rodent diet either supplemented with or without GRP. After 4 weeks of dietary treatments, half of the mice within each dietary group were subjected to 2.5% DSS treatment to induce colitis. Dietary GRP decreased DSS-induced body weight loss, disease activity index, and colon shortening. Glucoraphanin supplementation protected the colonic histological structure, suppressed inflammatory cytokines, interleukin (IL)-1β, IL-18, and tumor necrosis factor-α (TNF-α), and reduced macrophage infiltration in colonic tissues. Consistently, dietary GRP activated AMP-activated protein kinase (AMPK), peroxisome proliferator-activated receptor-gamma coactivator (PGC)-1α, and nuclear factor erythroid 2-related factor 2 (NRF2) pathways in the colonic tissues of DSS-treated mice, which was associated with increased mitochondrial DNA and decreased content of the oxidative product 8-hydroxydeoxyguanosine (8-OHDG), a nucleotide oxidative product of DNA. In conclusion, dietary GRP attenuated mitochondrial dysfunction, inflammatory response, and oxidative stress induced by DSS, suggesting that dietary GRP provides a dietary strategy to alleviate IBD symptoms.

## 1. Introduction

Inflammatory bowel diseases (IBDs), such as Crohn’s disease and ulcerative colitis, are chronic inflammatory disorders that affect the gastrointestinal tract, and their incidence and clinical severity have increased worldwide [1]. IBDs have a very complicated etiology, which is closely associated with immune system dysregulation (caused by genetic or environmental factors), impaired intestinal homeostasis, mitochondrial dysfunction, and dysbiosis of gut microbiota. Since the gastrointestinal tract is exposed to numerous commensal bacteria and opportunistic pathogens, defects in the intestinal mucosal barrier allow immunogenic factors to infiltrate into the mucosa, causing chronic inflammation [2,3].

A wide variety of dietary bioactive compounds, such as polyphenols, are known to have anti-inflammatory effects [4,5]. Grape seed extract, containing high amounts of flavonoids and phenolic compounds, reduced the risk of IBD through suppressing inflammation [6]. Similarly, Goji berries suppressed inflammation and ameliorated dextran sulfate sodium (DSS)-induced colitis in mice [7].

As the precursor of phytochemical sulforaphane (SFN), glucoraphanin (GRP) is enriched in cruciferous vegetables such as bok choy, broccoli, and cabbage. The inactive precursor GRP can be hydrolyzed by myrosinase to produce SFN. Heating broccoli inactivates myrosinase, and microbiota-produced myrosinase plays an important role in hydrolyzing GRP [8]. SFN has been tested in various disease models for its preventive effects on inflammation, liver injury and steatosis [9,10]. Previous work has shown that sulforaphane alleviated colitis severity in mice induced by DSS, supported by suppressed inflammatory cytokine expression, enhanced M2 macrophage polarization, and increased probiotic *Butyricicoccus* abundance [11,12,13].

SFN robustly activates the Kelch-like ECH-associated protein 1- Nuclear factor erythroid 2- related factor 2 (KEAP1-NRF2) pathway [14,15]. NRF2 is a transcription factor that plays an important role in redox homeostasis. In normal conditions, KEAP1 is constantly expressed and located in the cytoplasm, and it tethers the transcription factor NRF2 to repress its function [16]. When oxidation occurs, NRF2 is released and translocated into the nucleus, where it binds to antioxidant response elements (ARE) in the promoters of genes involved in cellular stress responses to initiate their expression [17]. Phase 2 and antioxidative enzymes play an important role in detoxifying harmful chemicals and intermediates, which serve as a key cellular defense mechanism against carcinogens [18]. Several cytoprotective phase 2 enzymes are upregulated by NRF2, including heme oxygenase-1 (HO-1), NAD(P)H: quinone oxidoreductase (NQO-1), and glutathione S-transferases (GSTs) [19,20]. These enzymes have anti-inflammatory and antioxidative activities [21]. The NRF2 pathway also decreases the production of pro-inflammatory cytokines and suppresses inflammation [22], while *Nrf2*-deficient mice have decreased phase 2 enzyme activities, such as GST and NQO-1 [23]. As a result, *Nrf2* knockout mice were more sensitive to DSS-induced colitis, demonstrating severe colitis symptoms, with increased pro-inflammatory cytokine levels and decreased antioxidative enzyme expression [24]. These data show that NRF2 is a potential therapeutic target for IBD prevention and treatment. In addition, the risk of colorectal cancer is dramatically increased in patients diagnosed with colitis [25]. SFN exhibits anticancer properties in cell culture and animal models through different pathways and mechanisms [26,27]. SFN can block HT29 human colon cancer cells in G_2_-M Phase and induce cell apoptosis [28].

The NRF2 pathway also protects against mitochondrial dysfunction in neurodegenerative diseases [29,30,31]. Mitochondrial dysfunction is also associated with IBD progression [32]. However, the effects of SFN/GRP on mitochondrial homeostasis in DSS-induced colitis have not been examined. Here, we hypothesized that broccoli-derived GRP attenuates DSS-induced colitis through activation of the NRF2 pathway, suppressing oxidative stress and inflammation while improving mitochondrial homeostasis.

## 2. Materials and Methods

### 2.1. Animals and Experimental Design

C57BL/6 mice were purchased from Jackson Laboratory (Bar Harbor, ME, USA). All animal procedures were approved by the Washington State University Institutional Animal Use and Care Committee (IAUCC). All mice were housed in a temperature-controlled room with a 12-h light/dark cycle. Mice had free access to food and water. Feed was changed daily to minimize the oxidation of functional compounds. Eight-week-old mice were randomly assigned to 2 groups, receiving either a control diet (AIN-93G purified diet, Harlan Laboratory) or a diet supplemented with glucoraphanin (600 ppm, Thorne Research) for 4 weeks. Then, mice in each dietary group were further randomly divided into two subgroups receiving no DSS, or 2.5% (*w*/*v*) DSS (Millipore, Billerica, MA, USA) for 9 days to induce colitis, followed by a 9-day recovery period with regular drinking water. Body weight was monitored daily during DSS treatment.

### 2.2. Assessment of Colitis Symptoms and Disease Activity Index

The disease activity index (DAI) score was assessed according to the previously described method [33]. Body weight loss (scored as 0–4), stool consistency (scored as 0–4), and fecal bleeding (scored as 0–4) were recorded daily during the DSS induction and recovery period. DAI score was recorded as the summation of body weight loss, stool consistency, and fecal bleeding.

### 2.3. Colon Sample Collection and Processing

At necropsy, mice were anesthetized with CO_2_ inhalation, followed by cervical dislocation. The colon section was dissected, and a 5 mm segment of the distal colon was fixed in 4% (*w*/*v*) paraformaldehyde (pH 7.0), processed, and embedded in paraffin for histological analysis. The remaining colon tissue was rinsed with PBS, frozen in liquid nitrogen, and stored at −80 °C for molecular and biochemical analysis.

### 2.4. Histological Evaluation

Histological evaluation was performed as previously described [7]. Paraffin-embedded distal colonic tissues were sectioned at 5 μm thickness, deparaffinized, and subjected to hematoxylin and eosin (H&E) staining. Histological examination and imaging were performed using a Lecia DM2000 LED light microscope (Leica Microsystems, Chicago, IL, USA). Each colonic section was scored blindly using previously published score criteria. The scores of crypt damage (0–4 scale), the severity of inflammation (0–3 scale), and depth of injury (0–3 scale) were recorded for each mouse. The pathobiological score is the summation of these scores on a scale from 0 to 10.

### 2.5. Immunoblotting Analyses

Powdered colonic samples were homogenized in a Precellys homogenizer (Rockville, MD, USA) with lysis buffer. The colonic protein extracts were separated by 10% SDS-polyacrylamide gel electrophoresis and transferred to nitrocellulose membranes. After blocking with 5% *w*/*v* bovine serum albumin, membranes were incubated overnight with the selected primary antibodies at 4 °C. The membranes were rinsed three times using Tris-buffered saline (TBS) with 0.5% Tween 20, followed by incubation with either IRDye 680 goat anti-mouse or IRDye 800CW goat anti-rabbit secondary antibodies (Li-Cor Biosciences, Lincoln, NE, USA). Finally, the bands were visualized and quantified using the Odyssey Infrared Imaging System and Image Studio Lite software (Li-Cor Biosciences, Lincoln, NE, USA). Band density was normalized to the β-tubulin content. Antibodies against interleukin (IL)-1β (#12242), TNF-α (#3707), phos-AMP-activated protein kinase (AMPK) (#2535), AMPK (#5831), NRF2 (#12721), OPA1 Mitochondrial Dynamin Like GTPase (OPA1) (#80471), dynamin-related protein 1 (DRP1) (#8570) and HO-1 (#5853) were purchased from Cell Signaling Technology (Beverly, MA, USA), and antibodies against superoxide dismutase (SOD) 1 (sc-17767) and xanthine oxidase (sc-398548) were purchased from Santa Cruz Biotechnology Inc. (Dallas, TX, USA). Antibody against peroxisome proliferator-activated receptor-gamma coactivator (PGC-1α) (#66369-I) was purchased from Protintech (Rosemont, IL, USA). The antibodies against β-tubulin (#E7) and IL-18 (CPTC-IL18) were obtained from the Developmental Studies Hybridoma Bank, University of Iowa (Iowa City, IA, USA). All antibodies were used at 1:1000, except NRF2, SOD1, and xanthine oxidase which were used at 1:500.

### 2.6. Immunohistochemical Staining

Deparaffinized and rehydrated colonic tissue sections were permeabilized in Tris-buffered saline (TBS) containing 0.3% Triton X-100 at room temperature for 30 min, boiled in sodium citrate buffer (pH 6.0) for 20 min, blocked with 10% goat serum (Vector Laboratories, Burlingame, CA, USA) in TBS (pH 7.4) for 1 h to reduce non-specific binding, then incubated overnight with primary antibodies against F4/80 (Bio-Rad Laboratories, Hercules, CA, USA, MCA497R, 1:200) or 8-hydroxydeoxyguanosine (8-OHDG) (Santa Cruz Biotechnology, sc-39387, 1:100) at 4 °C. Tissue sections were then incubated with biotinylated secondary antibodies (1:200, Vector Laboratories) at room temperature for 30 min. The positive cells were visualized using a Vectastain ABC and DAB peroxidase (HRP) substrate kit (Vector Laboratories), followed by hematoxylin counterstaining. Images were obtained using a Lecia DM2000 LED light microscope (Leica Microsystems).

### 2.7. Mitochondrial DNA Copy Number

Total DNA was isolated from colon tissue. Mitochondrial DNA (mtDNA) copy number was measured by quantitative PCR (qPCR) and normalized using nuclear 18S rRNA gene copies. Relative mtDNA copy number was calculated by using the 2^−ΔΔCT^ method.

### 2.8. Statistical Analysis

Data were analyzed as a complete randomized design using the General Linear Model and Statistical Analysis System (SAS Institute Inc., Cary, NC, USA). Unpaired Student’s *t*-test and two-way ANOVA were used for identifying the significant difference. Data were presented as mean ± standard error of the means (SEM). A significant difference was considered as *p* ≤ 0.05.

## 3. Results

### 3.1. Glucoraphanin Ameliorates the Symptoms of DSS-Induced Colitis

Dietary GRP supplementation reduced the body weight loss induced by DSS (Figure 1A), significantly improved fecal consistency and reduced blood in the stool (Figure 1B,C). Overall, dietary GRP improved the DAI score from day 9 to day 18 in mice challenged with DSS (Figure 1D). Glucoraphanin supplementation prevented colon length shortening in DSS-treated mice (Figure 1E,F). These data showed that GRP supplementation has protective effects against DSS-induced colitis.

### 3.2. Glucoraphanin Alleviates Histological Damage and Suppresses Inflammation in DSS-Induced Colitis

Histologically, DSS elicited notable distortion of crypts in the intestine, which was rescued by GRP supplementation (Figure 2A,B). F4/80 is a well-established marker for macrophages [34]. The F4/80-positive cells detected by IHC staining were decreased by dietary GRP in mice with DSS-induced colitis (Figure 3A,B), suggesting that GRP supplementation reduced macrophage infiltration in DSS-treated mice. Furthermore, GRP supplementation reduced the expression of inflammatory cytokines IL-1β, IL-18, and TNF-α (Figure 3C). Data suggested that GRP protects the colonic structure and prevents inflammatory response in DSS-induced colitis.

### 3.3. Glucoraphanin Activates AMPK and Mitochondrial Biogenesis in DSS-Induced Colitis

Hyperactivation of the inflammatory response is associated with DSS-induced colitis. AMPK is known for its anti-inflammatory effects [35]. GRP supplementation elevated AMPK activity as shown by increased AMPK phosphorylation (Figure 4A). There is a strong correlation between AMPK and PGC-1α [36,37]. As the major regulator of mitochondria biogenesis, PGC-1α expression was also increased by GRP intake in DSS-treated mice (Figure 4B). Furthermore, GRP supplementation increased mtDNA copy number in DSS-treated mice (Figure 4C). No difference was found in the protein expression of OPA1 which mediates mitochondrial fusion [38]. DRP1, which regulates mitochondrial fission [39], was significantly increased in DSS-treated mice, but suppressed by dietary GRP. These data suggested that GRP supplementation activates AMPK and upregulates mitochondrial biogenesis in DSS-challenged mice.

### 3.4. Glucoraphanin Supplementation Activates NRF2 and Alleviates Oxidative Stress

Oxidative stress plays a significant role in the etiology of colitis and other inflammatory diseases. NRF2 is the master regulator of antioxidative responses that is activated by SFN [40]. Consistently, NRF2 was elevated by GRP supplementation in mice regardless of DSS induction (Figure 5A). Its downstream target HO-1 was lower in DSS-treated mice compared with control mice without DSS induction but restored by GRP supplementation. The content of xanthine oxidase was higher in DSS-treated mice compared to CON mice supplemented with or without GRP (Figure 5A). However, supplementation of GRP did decrease the level of xanthine oxidase compared to non DSS-treated mice. No difference was observed in SOD1 protein content between CON and DSS-treated mice (Figure 5A). The levels of 8-OHDG, a marker for DNA damage, were markedly increased in DSS-treated mice (Figure 5BC) but decreased by GRP supplementation. The results suggested the protective effects of GRP against DSS-induced DNA damage.

## 4. Discussion

The incidence and prevalence of IBD are increasing across the world. North America and Europe have the highest incidence and prevalence of IBDs, which might be associated with the westernized diet [41]. However, the incidence of IBDs has also risen in developing countries in recent decades [1,42]. To prevent the relapse or improve the quality of life for IBD patients, various therapeutic approaches to IBD have been investigated, such as using anti-tumor necrosis factor (TNF) antibodies, Janus kinase inhibitors, and immunosuppressant drugs [43,44]. When patients received these drugs, they frequently experienced side effects such as asthma, diarrhea, and skin rash [45]. Natural bioactive compounds, enriched in foods and medicinal plants, provide an alternative for the treatment of chronic inflammatory diseases. Red raspberries reduced inflammation and protected mice from DSS-induced colitis through decreased immune cell infiltration, increased antioxidant enzyme expression, and enhanced intestinal barrier function [46]. Panaxynol extracted from American ginseng also showed beneficial effects on DSS-induced colitis, which exerted anti-inflammatory properties and prevented DNA damage in macrophages [47]. In a previous study, SFN administration prevented *H. pylori*-induced gastritis and suppressed non-steroidal anti-inflammatory drugs (NSAIDs)-induced oxidative stress [48]. Moreover, SFN inhibited histone deacetylase (HDAC) activity, which enhanced histone acetylation on the *bax* and *p21* promoters and their expression levels in prostate epithelial cells [49]. In DSS-induced colitis mice, dietary SFN supplementation decreased body weight loss [12], alleviated the severity of colitis induced by DSS, and prevented associated dysbiosis [11]. Consistently, we found that dietary GRP rescued the DSS-induced body weight loss, colon shortening, and decreased DAI score.

Inflammatory mediators and cytokines play a crucial role in IBD development and progression. IL-18 impairs the development and maturation of colonic goblet cells. The deletion of IL-18 conferred resistance against colitis. On the other hand, activation of IL-18 signaling exaggerated the progression of DSS-induced colitis associated with the depletion of goblet cells [50]. TNF-α and members of the IL family, such as IL-1, IL-6, IL-12, and IL-18, are produced by macrophages, dendritic cells, and epithelial cells, which promote inflammation [51,52]. Macrophages, neutrophils, and eosinophils were found to accumulate in DSS-induced acute colitis [53]. In this study, we found that GRP supplementation decreased macrophage infiltration in colonic tissue and downregulated inflammatory cytokines. Thus, dietary GRP protects against DSS-induced colitis through the suppression of the inflammatory response and immune cell infiltration.

AMPK is a key regulator of energy homeostasis and enhances intestinal epithelial differentiation and barrier function [54]. Metformin, an indirect AMPK activator, improves ileal epithelial barrier function in IL-10 knockout mice [55] and rescues intestinal barrier dysfunction induced by lipopolysaccharide [56]. AMPK can be activated by bioactive compounds such as resveratrol and polyphenol-rich foods such as purple potato and raspberry [57,58,59]. Our data showed that dietary GRP increased AMPK phosphorylation in DSS-induced mice. The PGC-1α level, a well-known AMPK downstream target [36], was also elevated in GRP-supplemented mice, indicating that GRP protected against colitis partially through stimulating the AMPK/PGC-1α pathway.

Mitochondrial function is indispensable for proper intestinal stem cell differentiation and function [60,61]; PGC-1α is a master regulator of mitochondrial biogenesis and respiration [62]. In ulcerative colitis patients, the intestine shows reduced mitochondrial oxidative phosphorylation (OXPHOS) [63,64] and decreased expression of genes involved in mitochondrial function [65]. In this study, dietary GRP increased PGC-1α, suggesting improved mitochondria biogenesis in DSS-induced colitis. In agreement, mtDNA copy number was significantly decreased in mice treated with DSS, which was restored by GRP supplementation. Mitochondrial fission is mediated by the dynamin-related GTPase, DRP1, which cooperates with other mitochondrial outer membrane proteins, such as Fis1 and Mff in mitochondrial fission [66]. Sulforaphane enhances the hyperfusion of mitochondria and impairs fission machinery in an NRF2-independent manner in human retinal pigment epithelial cells [67]. To remove dysfunctional mitochondria, fusion and fission are tightly regulated, in which fission is responsible for mitochondrial autophagy [68]. A recent study reported that DRP1 levels were increased in DSS-treated mice [69]. However, the underlying mechanisms of mitochondrial fusion and fission events in IBD are not fully understood. Here, we found that DRP1 was upregulated in DSS-treated mice, which was mitigated by GRP supplementation, without the involvement of OPA1. Our data suggested that GRP maintained mitochondrial homeostasis in DSS-induced colitis through suppression of mitochondrial fission.

Besides inflammation, oxidative stress is another major pathogenic factor in inflamed tissue; reactive oxygen species (ROS) associated with oxidative stress produces oxidation products and causes DNA damage [70]. The NRF2 pathway plays an important role in maintaining redox homeostasis [71]. When NRF2 is inactivated, it is trapped and degraded in the cytoplasm. The activated NRF2 translocates to the nuclei and initiates the expression of its target genes, including antioxidative and anti-inflammatory mediators and enzymes [72]. The endogenous antioxidant defense mechanism consists of several phase II detoxification enzymes, including GST and HO-1 [73,74]. The expression of these antioxidant enzymes is regulated by promoters containing ARE, which is bound by the NRF2 transcription factor [75]. In agreement with previous studies, we found that NRF2 and xanthine oxidase were highly elevated in GRP-supplemented DSS-treated mice. HO-1, as a rate-limiting enzyme in heme catabolism, possesses a wide spectrum of protective properties including antioxidative, anti-inflammatory, and antiapoptotic effects [76]. However, either an increase or decrease of HO-1 can be found in mice treated with DSS [77,78]. Our data showed increased HO-1 by GRP supplementation, consistent with increased NRF2. GRP supplementation did not change SOD1 levels, suggesting the preventative effect of GRP against oxidative stress might be through other mechanisms. DNA damage is associated with the initiation and promotion of carcinogenesis. 8-OHdG is one of the major products generated by free-radical-induced oxidation that causes DNA damage [79]. Consistent with the enhanced NRF2 and antioxidant enzymes, GRP supplementation decreased 8-OHdG levels in the colonic tissue of DSS-treated mice.

Inflammatory bowel diseases are constantly associated with dysbiosis of the gut microbiota. Dietary GRP was previously found to modulate gut microbiota in a high-fat diet mouse model. Body weight and fat mass were significantly decreased by GRP supplementation, associated with an increased abundance of *Akkermansia* and *Alloprevotella* [80]. In a DSS-induced colitis mouse model, SFN was reported to increase the abundance of a butyrate-producing bacteria, *Butyricicoccus,* and improve the symptoms of colitis [11]. As the precursor of SFN, GRP can be partially hydrolyzed by gut microbiota. Additional studies are required to evaluate the effects of GRP on IBD symptoms and alterations of the gut microbiota.

## 5. Conclusions

The results demonstrated that dietary GRP ameliorated the pathological score and colon shortening induced by DSS. The supplementation of GRP reduced the inflammatory response, activated NRF2, and alleviated oxidative stress in the colon of DSS-treated mice. Mitochondrial homeostasis was maintained by dietary GRP in a mouse model of DSS-induced colitis. Therefore, GRP provides an alternative strategy for alleviating IBD symptoms.

## Figures and Tables

**Figure 1 antioxidants-11-02404-f001:**
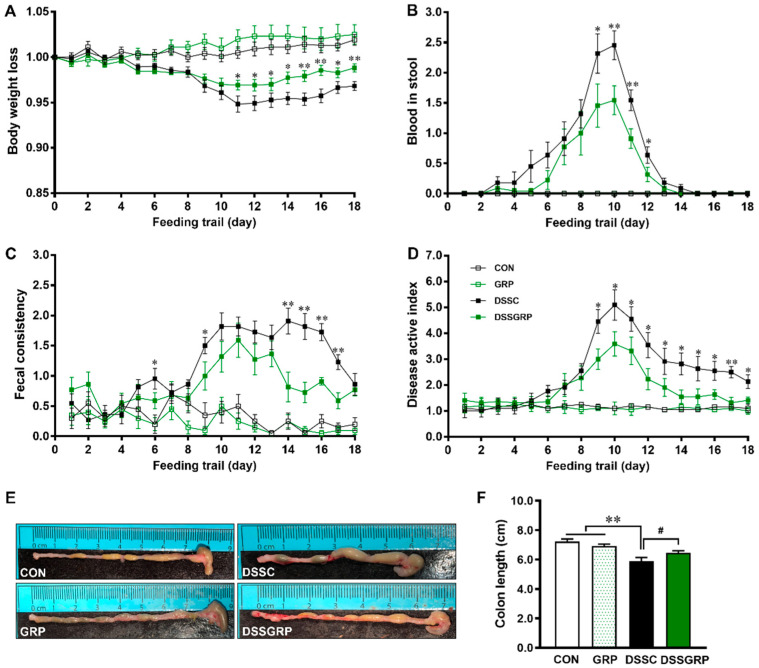
GRP supplementation suppresses disease symptoms in DSS-induced colitis mice. (**A**) Body weight loss. (**B**) Blood in stool. (**C**) Fecal consistency. (**D**) Disease activity index. (**E**) The representative images of the dissected colon. (**F**) Colon length. CON: Control, GRP: Sulforaphane glucosinolate, DSSC: DSS-treated control mice, DSSGRP: DSS-treated mice with GRP supplementation. Mean ± SEM, *n* = 10. ^#^
*p* < 0.10, * *p* ≤ 0.05, ** *p* ≤ 0.01.

**Figure 2 antioxidants-11-02404-f002:**
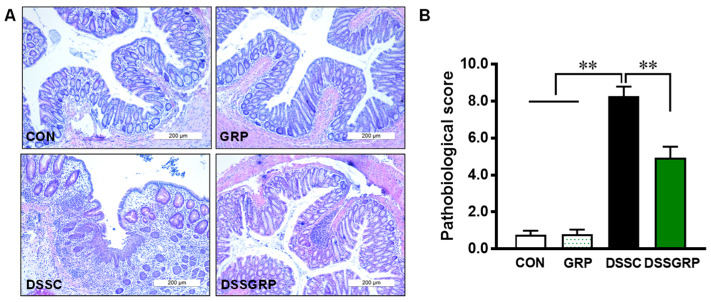
GRP supplementation decreases pathological scores in the colonic tissues of mice. (**A**) Representative hematoxylin and eosin staining. (**B**) The pathological score of distal colonic tissues. CON: Control, GRP: Sulforaphane glucosinolate, DSSC: DSS-treated control mice, DSSGRP: DSS-treated mice with GRP supplementation. *n* = 8–9 mice per group. ** *p* ≤ 0.01.

**Figure 3 antioxidants-11-02404-f003:**
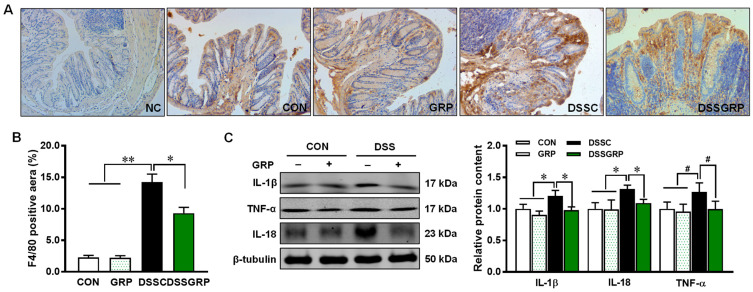
GRP supplementation inhibits macrophage infiltration and suppresses inflammation in the colonic tissues of mice with DSS-induced colitis. (**A**) Representative images of F4/80 staining. (**B**) The statistics of F4/80 staining. (**C**) Relative protein contents of IL-1β, TNF-α, and IL18. NC: Negative control, CON: Control, GRP: Sulforaphane glucosinolate, DSSC: DSS-treated control mice, DSSGRP: DSS-treated mice with GRP supplementation. *n* = 8–9 mice per group. ^#^
*p* < 0.10, * *p* ≤ 0.05, ** *p* ≤ 0.01.

**Figure 4 antioxidants-11-02404-f004:**
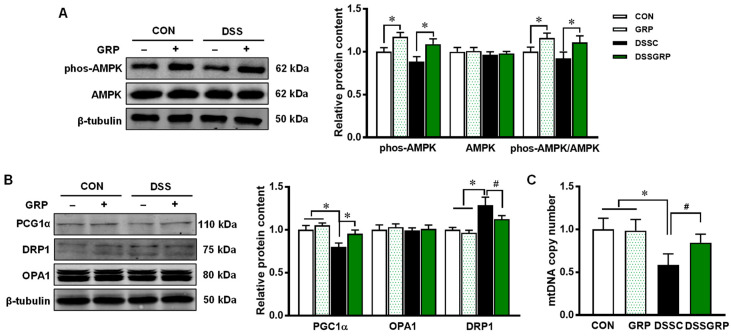
GRP supplementation activates AMPK and increases mitochondrial biogenesis. (**A**) Relative protein contents of phospho-AMPK and AMPK. (**B**) Relative protein contents of PGC-1α, DRP1, and OPA1. (**C**) Mitochondrial DNA copy number. CON: Control, GRP: Sulforaphane glucosinolate, DSSC: DSS-treated control mice, DSSGRP: DSS-treated mice with GRP supplementation. *n* = 8–9 mice per group. ^#^
*p* < 0.10, * *p* ≤ 0.05.

**Figure 5 antioxidants-11-02404-f005:**
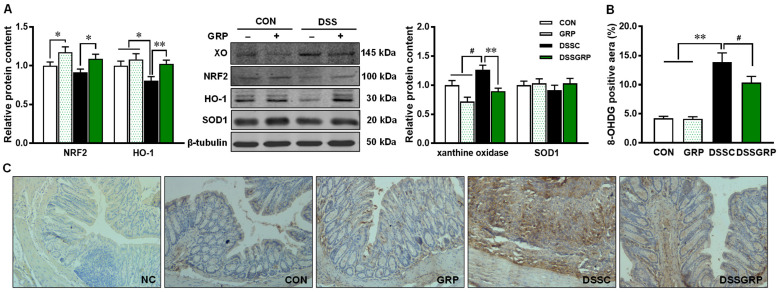
GRP supplementation activates NRF2 signaling and attenuates oxidative stress in the colonic tissues of mice with DSS-induced colitis. (**A**) Relative protein contents of NRF2, HO-1, xanthine oxidase (XO), and SOD1. (**B**) 8-OHdG staining statistics. (**C**) Representative images of 8-OHdG staining. NC: Negative control, CON: Control, GRP: Sulforaphane glucosinolate, DSSC: DSS-treated control mice, DSSGRP: DSS-treated mice with GRP supplementation. *n* = 8–9 mice per group. ^#^
*p* < 0.10, * *p* ≤ 0.05, ** *p* ≤ 0.01.

## Data Availability

The data presented in this study are available in the article.

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
