# Peer review of "Broccoli-Derived Glucoraphanin Activates AMPK/PGC1α/NRF2 Pathway and Ameliorates Dextran-Sulphate-Sodium-Induced Colitis in Mice"

_antioxidants, 2022, doi:10.3390/antiox11122404_

Round 1
Reviewer 1 Report
The article entitled “Broccoli-derived glucoraphanin activates AMPK/PGC1α/NRF2 2 pathway and ameliorates dextran sulphate sodium -induced co- 3 litis in mice” Is focused on the effect of glucoraphanin both as anti-inflammatory and anti oxidative stress molecules.
In general the article is well written and the experiments are well organized, however the author have to explain the sentence that these molecules can activate Nrf2.
As the same authors have affirmed, Nrf2 is a nuclear factor and its activation can not be detected simply by its cytoplasmatic level, instead by the level of this protein in the nucleus after its activation or by other assay (i.e. luciferase assay). To maintain their conclusion the authors should make these analysis in order to assess Nrf2 activation.
Author Response
The article entitled “Broccoli-derived glucoraphanin activates AMPK/PGC1α/NRF2 2 pathway and ameliorates dextran sulphate sodium -induced co- 3 litis in mice” Is focused on the effect of glucoraphanin both as anti-inflammatory and anti oxidative stress molecules.
In general the article is well written and the experiments are well organized, however the author have to explain the sentence that these molecules can activate Nrf2.
As the same authors have affirmed, Nrf2 is a nuclear factor and its activation can not be detected simply by its cytoplasmatic level, instead by the level of this protein in the nucleus after its activation or by other assay (i.e. luciferase assay). To maintain their conclusion the authors should make these analysis in order to assess Nrf2 activation.
Response: Thanks for your valuable comments. Based on available studies, NRF2 regulation is mainly through KEAP1-mediated degradation. Oxidative stress suppresses KEAP1-mediated NRF2 ubiquitin degradation, leading to NRF2 accumulation which enters the nucleus to initiate gene expression. Up to now, there is no evidence supporting the regulated NRF2 nuclear translocation, and NRF2 translocation appears to be spontaneous through its nuclear localization signaling. Therefore, nuclear translocation is correlated with the accumulation of NRF2, or the total content of NRF2, which we measured through Western blotting. Therefore, measurement of its nuclear localization may not be necessary. Thanks again!
Reviewer 2 Report
This article describes the effects of glucoraphanin on dextran sulphate sodium-induced colitis in mice and the anti-oxidant action mechanism of glucoraphanin.
The experimental design seems to be exquisite, and the results obtained in each experiment are consistent.
However, the following points would be considered to improve this article.
Minor point
1) Some results showed that the effect of glucoraphanin was small.
Moreover, SOD1 protein was not increased by glucoraphanin in Fig.5B.
Reviewer recommends to explain the reason why authors obtained these results in
the discussion section.
2) The expression style should be unified.
In “References” section
Line 346 : The Journal of Nutritional Biochemistry → J Nutr Biochem
Line 357 : The Journal of nutritional biochemistry → J Nutr Biochem
Line 374 : Archives of dermatological research → Arch Dermatol Res
Line 380 : Proceedings of the National Academy of Sciences
→ Proc Natl Acad Sci USA
Line 390 : Frontiers in cell and developmental biology → Front Cell Dev Biol
Line 401 : Proceedings of the National Academy of Sciences
→ Proc Natl Acad Sci USA
Line 411 : Nature Reviews Gastroenterology & Hepatology
→ Nat Rev Gastroenterol Hepatol
Line 417 : Journal of Crohn's and Colitis → J Crohns Colitis
Line 418 : The Journal of nutritional biochemistry → J Nutr Biochem
Line 430 : The American journal of pathology → Am J Pathol
Line 443 : The Journal of nutritional biochemistry → J Nutr Biochem
Line 445 : Proceedings of the National Academy of Sciences
→ Proc Natl Acad Sci USA
Line 462 : American Journal of Physiology-Gastrointestinal and Liver Physiology
→ Am J Physiol Gastrointest Liver Physiol
Line 468 : The EMBO journal → EMBO J
Line 470 : Cellular and molecular gastroenterology and hepatology
→ Cell Mol Gastroenterol Hepatol
Line 483 : The Journal of nutritional biochemistry → J Nutr Biochem
Line 486 : Frontiers in pharmacology → Front Pharmacol
Line 489 : International Immunopharmacology → Int Immunopharmacol
Line 492 : Journal of pediatric gastroenterology and nutrition
→ J Pediatr Gastroenterol Nutr
Line 495 : Journal of environmental science and health Part C
→ J Environ Sci Health C Environ Carcinog Ecotoxicol Rev
3) The expression style should be unified (Another version)
What is “-“ in “M.-T.” or “A.-N.” ?
Ex) Line 382 : Khor, T.O.; Huang, M.-T.; Kwon, K.H.; Chan, J.Y.;
Reddy, B.S.; Kong, A.-N.
4) Insufficient description
In “References” section
Line 435 : 2001, 1, 1-11. → 2001, 1, 3.
Line 448 : 2015, 1, 1-11. → 2015, 1, 15063.
Author Response
This article describes the effects of glucoraphanin on dextran sulphate sodium-induced colitis in mice and the anti-oxidant action mechanism of glucoraphanin. The experimental design seems to be exquisite, and the results obtained in each experiment are consistent.
However, the following points would be considered to improve this article.
Minor point
- Some results showed that the effect of glucoraphanin was small. Moreover, SOD1 protein was not increased by glucoraphanin in Fig.5B. Reviewer recommends to explain the reason why authors obtained these results in the discussion section.
Response: Thanks for your valuable comments. We agree to your comments. SOD1 has been analyzed as an important antioxidant enzyme response to oxidative stress. The explanation has been added to the manuscript.
- The expression style should be unified.
In “References” section
Line 346 : The Journal of Nutritional Biochemistry → J Nutr Biochem
Line 357 : The Journal of nutritional biochemistry → J Nutr Biochem
Line 374 : Archives of dermatological research → Arch Dermatol Res
Line 380 : Proceedings of the National Academy of Sciences→ Proc Natl Acad Sci USA
Line 390 : Frontiers in cell and developmental biology → Front Cell Dev Biol
Line 401 : Proceedings of the National Academy of Sciences→ Proc Natl Acad Sci USA
Line 411 : Nature Reviews Gastroenterology & Hepatology→ Nat Rev Gastroenterol Hepatol
Line 417 : Journal of Crohn's and Colitis → J Crohns Colitis
Line 418 : The Journal of nutritional biochemistry → J Nutr Biochem
Line 430 : The American journal of pathology → Am J Pathol
Line 443 : The Journal of nutritional biochemistry → J Nutr Biochem
Line 445 : Proceedings of the National Academy of Sciences→ Proc Natl Acad Sci USA
Line 462 : American Journal of Physiology-Gastrointestinal and Liver Physiology→ Am J Physiol Gastrointest Liver Physiol
Line 468 : The EMBO journal → EMBO J
Line 470 :Cellular and molecular gastroenterology and hepatology → Cell Mol Gastroenterol Hepatol
Line 483 : The Journal of nutritional biochemistry → J Nutr Biochem
Line 486 : Frontiers in pharmacology → Front Pharmacol
Line 489 : International Immunopharmacology → Int Immunopharmacol
Line 492 : Journal of pediatric gastroenterology and nutrition → J Pediatr Gastroenterol Nutr
Line 495: Journal of environmental science and health Part → J Environ Sci Health C Environ Carcinog Ecotoxicol Rev
Response: Revised per suggestion.
- The expression style should be unified (Another version)
What is “-“ in “M.-T.” or “A.-N.” ?
Ex) Line 382 : Khor, T.O.; Huang, M.-T.; Kwon, K.H.; Chan, J.Y.; Reddy, B.S.; Kong, A.-N.
Response: Thanks for your comments. We have corrected the author’s name.
- Insufficient description
In “References” section
Line 435 : 2001, 1, 1-11. → 2001, 1, 3.
Line 448 : 2015, 1, 1-11. → 2015, 1, 15063.
Response: Thanks for your comments. We have changed the description.
Reviewer 3 Report
This is an interesting manuscript evaluating the effect GRP supplementation in preventing mitochondrial dysfunction and oxidative stress in an acute colitis mouse model.
Although the manuscript is interesting and generally well written is presents some flaws that must be resolved. In particular:
Lines 54-71:it deserves to be added that Sulforaphane can also improve cancer sensitivity to chemotherapeutics modulating NRF2/KEAP1 signaling (see PMID: 35901941, 36295538). This is an important point to add because it can further highlight the results obtained by the authors and underline the multifaceted role of this compound.
Authors should report the number of mice analysed in each experiment
2.5. Immunoblotting analyses: dilution and product code of primary antibodies must been reported
2.6. Immunohistochemical staining: dilution and product code of primary antibodies must been reported
Figure 2A, 3A and 5C: images must be improved because resolution is very low and they get blurry when zoomed in. Negative control in Figure 3A and 5C must be shown
A higher magnification in IHC figures should be shown
Figures showing images of western blot must report the molecular weights of each protein analysed
Author Response
This is an interesting manuscript evaluating the effect GRP supplementation in preventing mitochondrial dysfunction and oxidative stress in an acute colitis mouse model. Although the manuscript is interesting and generally well written is presents some flaws that must be resolved. In particular:
Lines 54-71:it deserves to be added that Sulforaphane can also improve cancer sensitivity to chemotherapeutics modulating NRF2/KEAP1 signaling (see PMID: 35901941, 36295538). This is an important point to add because it can further highlight the results obtained by the authors and underline the multifaceted role of this compound.
Response: Thank you very much for your suggestion. We agree that adding extra information about the anticancer properties of sulforaphane can help to understand our results better. The manuscript has been revised accordingly.
Authors should report the number of mice analyzed in each experiment
Response: Thanks for your comment. The sample size has been added.
2.5. Immunoblotting analyses: dilution and product code of primary antibodies must been reported
Response: Thanks for your comment. The Cat# and dilution of primary antibodies have been added.
2.6. Immunohistochemical staining: dilution and product code of primary antibodies must been reported
Response: Thanks for your comment. The Cat# and dilution of primary antibodies have been added.
Figure 2A, 3A and 5C: images must be improved because resolution is very low and they get blurry when zoomed in. Negative control in Figure 3A and 5C must be shown
A higher magnification in IHC figures should be shown
Response: Thanks for your comment. The higher magnification figures have been used including negative control.
Figures showing images of western blot must report the molecular weights of each protein analysed
Response: Thanks for your comment. The Cat# and dilution of primary antibodies have been added. The molecular weight of each protein has been added.
Round 2
Reviewer 1 Report
The answer of the authors is partially sufficient to answer to my question since the increase of protein expression of a nuclear factor should be correlated to its nuclear localization or activation as it is reported in different literature articles. In other word it the activation of Nrf2 is supported by the increase of HO-1 expression but to assert that "nuclear translocation is correlated with the accumulation of Nrf2, or the total content of Nrf2, which we measured through Western blotting" at least the authors should report some References. In fact the analysis of Nrf2 is relative only to the cytoplasmatic Nrf2 since in the Western blotting they use as house-keeping protein B-tubulin which is not a nuclear but a cytoplasmatic house-keeping protein.
Author Response
Response:
Thank you for your comments. Nrf2 has a relatively high turnover rate. Nrf2 is ubiquitinated by the KEAP1-based ubiquitin ligase complex, which rapidly undergoes proteasomal degradation [1], while Nrf2 is relatively stable in the absence of KEAP1. The inhibition of proteasomal degradation stabilizes Nrf2 and allows newly synthesized Nrf2 to translocate to the nucleus. It was observed that Nrf2 was principally found in the nucleus instead of cytosolic fractions in both HepG2 and H4IIEC3 cells. When the cell was treated with the phenolic antioxidant (tBHQ), Nrf2 in whole cell lysate and nuclear extracts fractions both increased without the change of cytoplasmic Nrf2 [2]. From previous studies, the whole cell lysate also can be used to analyze the Nrf2 accumulation (total Nrf2) [3,4]. These data suggest that total Nrf2 abundance is correlated with the Nrf2 accumulation in the nucleus.
- Itoh, K.; Wakabayashi, N.; Katoh, Y.; Ishii, T.; O'Connor, T.; Yamamoto, M. Keap1 regulates both cytoplasmic‐nuclear shuttling and degradation of Nrf2 in response to electrophiles. Genes to cells 2003, 8, 379-391.
- Nguyen, T.; Sherratt, P.J.; Nioi, P.; Yang, C.S.; Pickett, C.B. Nrf2 controls constitutive and inducible expression of ARE-driven genes through a dynamic pathway involving nucleocytoplasmic shuttling by Keap1. Journal of Biological Chemistry 2005, 280, 32485-32492.
- Li, Y.; Paonessa, J.D.; Zhang, Y. Mechanism of chemical activation of Nrf2. PloS one 2012, 7, e35122.
- Toyama, T.; Shinkai, Y.; Yasutake, A.; Uchida, K.; Yamamoto, M.; Kumagai, Y. Isothiocyanates reduce mercury accumulation via an Nrf2-dependent mechanism during exposure of mice to methylmercury. Environmental health perspectives 2011, 119, 1117-1122.
Reviewer 3 Report
the manuscript has been significantly improved and can be accepted in the present form.
Author Response
Response: We really appreciate your valuable comments.